# The Characteristics of Acoustic Emissions Due to Gas Leaks in Circular Cylinders: A Theoretical and Experimental Investigation

Kwang Bok Kim, Jun-Hee Kim, Je-Eon Jin, Hae-Jin Kim, Chang-Il Kim, Bong Ki Kim and Jun-Gill Kang *

Rm #306, Integrity Diagnostics Korea (IDK), IT Venture Town, 35, Techno 9-ro, Yuseong-gu, Daejeon 34027, Republic of Korea
* Correspondence: jgkang@cnu.ac.kr

**Abstract:** An acoustic emission (AE) is caused by the sudden release of energy by a material as a result of material degradation related to deformations, cracks, or faults within a solid. The same situation also occurs in leaks caused by turbulence in the fluid around the leak. In this study, analytical modeling for an AE due to leakage through a circular pinhole in a gas storage cylinder was performed. The displacement fields responsible for AEs, excited by the concentrated force (CF) associated with the turbulent flow though the pinhole, were derived by solving the Navier–Lamé equation. The CF as an excitation source was formulated in terms of a fluctuating Reynolds stress (FRS) and spatial Green's function. In particular, a series of experiments were conducted under different operating conditions to explore the characteristics of the AE signals due to leak in a gas cylinder. Finally, the simulation and experimental results were compared to verify the accuracy of the simulation results.

**Keywords:** acoustic emission; leakage; fluctuating Reynolds stress; Green's function; simulation

## 1. Introduction

Reliable maintenance of high-pressure gas vessels, especially those containing fuel gases, has increased exponentially over the past decade, because the gases are extremely hazardous under conditions that can cause them to ignite. The most important way to prevent disasters caused by gas leak fires is to detect leaks and their leak location and make subsequent repairs. There are several pipeline leak detection methods being used to detect and locate leaks in a limited area of pipeline [1]. These include flow rate monitoring [2], pressure fluctuation analysis [3], fiber optic sensing [4], acoustic sensing [5–8], and infrared imaging [9]. Among the various methods, acoustic methods have proven to be very effective in detecting leaks by eliminating disturbing signals through time and frequency domain analysis and characterization of acoustic emission parameters [10–12].

Acoustic emissions (AEs) are elastic or stress waves generated by a material as the result of a sudden release of energy from a localized source within the solid. As well as the material degradation related to deformation and crack and fracture development, and fluid flow through the hole or crack created in the container system also generate AEs. When a fluid leak occurs due to the pressure difference between the inside and outside of the container, the fluid ejected from the leak hole can be characterized by Lighthill's stress [13]. In the turbulence flow, a fluctuating Reynolds stress (FRS), corresponding to the interaction between the fluid and the solid wall, is dominant in Lighthill's stress. The Reynolds stress is associated with a direction perpendicular to the wall and, consequently, leads to the excitation of AEs that propagate through the container.

Many experimental investigations have been conducted to characterize the AE caused by gas leaks. Pollock et al. investigated the AE signal as a function of internal pressure, hole cross-sectional area, and leak rate [14]. They found that a low frequency sensor (30 kHz) is suitable for detecting small leaks and that the signal amplitude is nearly linear

with internal pressure. Yoshida et al. investigated AEs during gas leaks from pipes with straight and stepwise pinholes in the 100–1200 kHz frequency range [15]. For the straight pinhole, they found that the mean amplitude of the AE signal increased gradually as the pressure of the outgoing air increased, with no variation until 0.3 MPa. For stepwise pinholes, the results were dependent on the shallow depth: compared to the straight pinhole case, shorter shallow depths showed low mean amplitudes with similar pressure dependence, while longer shallow depths showed variation in the mean amplitude in the range of 0.11–0.15 MPa. Laodeno et al. also identified the effect of the hole geometry on the amplitude and peak frequency of AE due to air leakage [16]. Three types of pinholes used in the leakage test, namely straight-type, stepwise-type and cone-type, gave different results. Mostafapou et al. measured AE signals generated by pipe vibrations due to gas leaks and used the Fast Fourier Transform (FFT) to determine the resonant frequency of the observed signals [17]. The observed frequencies were confirmed by applying Donnell's non-linear theory.

Previously, we presented a mathematical model for AEs generated by point source (PS) in cylindrical structures [18]. The PS as an internal defect was represented by a concentrated force (CF) with both spatial and temporal characteristics. In cylindrical coordinates, the CF generates three scalar potentials responsible for one compressional (P) wave and two shear waves (horizontally and vertically polarized; SH and SV, respectively), which we designated as the concentrated force-incorporated potentials (CFIPs). The radial, tangential and axial displacements were solved by introducing CFIPs to the Navier–Lamé equation based on the model proposed by Morse and Feshbach [19]. To the best of our knowledge, no theoretical work on AE excited by gas leakage in cylindrical geometries has been presented in the literature. The main objective of this work is to present a mathematical formula applicable to AEs due to a pinhole leakage in cylindrical shells. In this study, we derived CFIPs responsible for the FRS and analyzed the displacements associated with P, SH and SV waves on the outer surface of the cylindrical shell. For completeness, an experimental investigation was also carried out with the aim of characterizing the AE signals due to pinhole gas leakages from a cylinder. The artificial leak system was constructed with a commercial $N_2$ gas cylinder with a pinhole-type leakage source plugged into the cylinder's wall. The AE parameters such as frequency, mean amplitude and RMS were measured as a function of pressure and pinhole size. The resonant frequency distribution was obtained by decomposing the observed AE signal by the FFT method. In addition, the observed AE signal due to gas leakage through the pinhole was modeled by the theoretical formula. This paper provides the overall process of the generation, propagation, and reception of the AE signal due to leakage by establishing a mathematical model as well as experimental data.

## 2. Analytical Model

### 2.1. Fluctuating Reynolds Stress as Point Source

First, let us define the force $f$ generated by gas leakage. When a leak occurs through a small hole, turbulence is generated as the fluid upstream of the pinhole accelerates toward the hole and can be continuously affected by real physical boundaries such as solid walls. In fact, turbulent fluctuations, known as FRS, contribute to the mean motion of the fluid, meaning that FRS is actually an additional momentum flux [13]. If the momentum in the $x_i$ direction crosses a unit surface area in the $x_j$ direction, the net flux of $x_i$-momentum with a negative sign can be expressed as [13]

$$T'_{ij} = -\rho v'_i v'_j, \tag{1}$$

where the primes represent fluctuating states and $v$ is the velocity along a given direction. In fact, the FRS in Equation (1) is for instantaneous fluctuations. Therefore, it must be averaged over a period of time, as follows

$$\overline{T'_{ij}} = -\rho \overline{v'_i v'_j}. \tag{2}$$

The FRS on the hole wall is the most effective source of AE generated by leakage. For turbulent flow with a radial velocity of $v_r$ through a circular hole, FRS occurs along the vertical axes (axial and tangential axes), as shown in Figure 1a. The CF generated by the gas leaking through the pinhole can be written as follows

$$P_z = -\rho \frac{\partial \left( \overline{v_r' v_z'} \right)}{\partial x_z} \bigg|_{\text{wall}} \quad \text{and} \quad P_\theta = -\rho \frac{\partial \left( \overline{v_r' v_\theta'} \right)}{\partial x_\theta} \bigg|_{\text{wall}}, \tag{3}$$

where $P_z = P_\theta \equiv P_w$. In this study, we determined the FRS from the reported date [20].

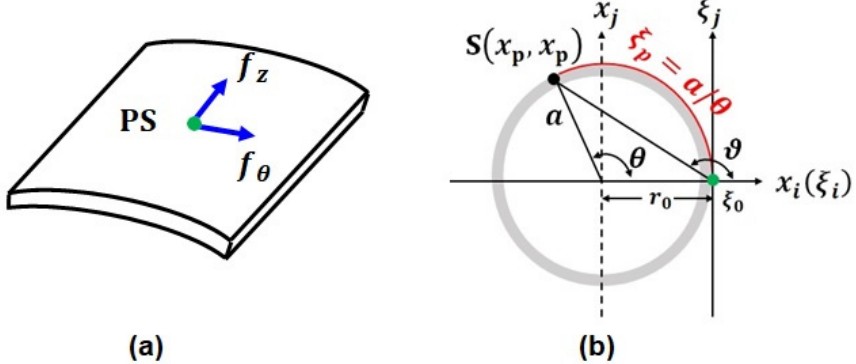

**Figure 1.** (**a**) Two forms of the PS vector along the axial and tangential directions caused by a pinhole leakage, and (**b**) a radial cross-section containing the PS and a given point S, in which the red line represents the arc connecting PS and S.

Figure A1 (in Appendix B) shows the plot of the Reynolds stress quantity (RSQ) vs. the wall-distance divided by layer distance ($\delta$). In the figure, RSQ is defined as [20]

$$RSQ = -20 \frac{\overline{u'v'}}{U^2}. \tag{4}$$

where $\overline{u'v'}$ is assumed to be the mean value of the product of $v_r' v_w'$, and $U$ is the mean velocity of $v_r'$. The experimental data is well fitted by a fifth-order polynomial

$$RSQ = 0.028 - 0.013\left(\frac{y}{\delta}\right) + 0.013\left(\frac{y}{\delta}\right)^2 - 0.155\left(\frac{y}{\delta}\right)^3 + 0.193\left(\frac{y}{\delta}\right)^4 - 0.065\left(\frac{y}{\delta}\right)^5$$

Substituting RSQ into Equation (3), we obtained

$$P_w = 6.62 \times 10^{-4} \rho U^2. \tag{5}$$

According to the mathematical model of hole leakage [21–23], the characteristics of gas flow, such as mass flow rate ($Q$) and mean flow velocity in the leak hole, can be divided into sonic flow and subsonic flow according to the critical pressure ratio (CPR)

$$CPR = \frac{p_a}{p_{cr}} = \left(\frac{2}{\gamma+1}\right)^{\frac{\gamma}{\gamma-1}}, \tag{6}$$

where $p_a$ is the atmosphere pressure of the surrounding environment, $p_{cr}$ is the critical pressure when the gas in the leakage section changes from subsonic to sonic flow, and $\gamma$ is

the isentropic index. When the gas pressure in the gas container ($p_0$) is lower than $p_{cr}$, the gas flow is in subsonic flow state, and the mass flow rate is given by

$$Q = C_D A_{lh} P_0 \sqrt{\frac{2\gamma}{\gamma - 1} \frac{M}{ZRT_2} \left[ \left(\frac{P_a}{P_0}\right)^{\frac{2}{\gamma}} - \left(\frac{P_a}{P_0}\right)^{\frac{\gamma+1}{\gamma}} \right]}, \tag{7}$$

where $C_D$ is the flow correction factor of the leakage hole (0.6~1.0), $A_{lh}$ is the cross-sectional area of the hole, and $Z$ is the compressibility factor of the gas. When $p_0 \geq p_{cr}$, the gas flow is in a sonic flow state. In this case,

$$Q = C_D A_{lh} p_0 \sqrt{\frac{\gamma M}{ZRT_0} \left(\frac{2}{\gamma + 1}\right)^{\frac{\gamma+1}{\gamma-1}}}. \tag{8}$$

Since the mass flow rate is the mass of the gas passing through the hole region over a period of time, the mean flow velocity can be written as

$$U = \frac{V_r}{A} = \frac{4V_r}{D^2 \pi} = \frac{4Q/\rho}{D^2 \pi}, \tag{9}$$

where $D$ is the diameter of the leakage hole.

### 2.2. Displacement Fields

The NL equation is the fundamental equation governing wave motion in elastic and homogeneous media. If the media is subjected to a non-equilibrium local force $f$, the NL equation can be written in vector form [24] as

$$(\lambda + 2\mu) \nabla(\nabla \cdot u) - \mu \nabla \times \nabla \times u + f = \rho \frac{\partial^2 u}{\partial t^2}, \tag{10}$$

where $u$ is the displacement vector, $\lambda$ and $\mu$ are Lamé constants, and $\rho$ is the density of the media. The displacement field in cylindrical coordinates is specified by three potentials: the scalar potential $\Phi$ for the P wave, and two vector potentials, $X\hat{e}_z$ for the SH wave and $\Psi\hat{e}_z$ for the SV wave. Previously, we adapted the model proposed by Morse and Freshbach [19], expressed by

$$u = \nabla\Phi + \nabla \times (X\hat{e}_z) + a\nabla \times \nabla \times (\Psi\hat{e}_z), \tag{11}$$

because the three components were easily separated. The displacement vector can be described by the displacement components in the $(r, \theta, z)$ coordinates

$$u = u_r \hat{r} + u_\theta \hat{\theta} + u_z \hat{z}, \tag{12}$$

where

$$u_r = \frac{\partial \Phi}{\partial r} + \frac{1}{r}\frac{\partial X}{\partial \theta} + a\frac{\partial^2 \Psi}{\partial r \partial z}, \tag{13}$$

$$u_\theta = \frac{1}{r}\frac{\partial \Phi}{\partial \theta} - \frac{\partial X}{\partial r} + \frac{a}{r}\frac{\partial^2 \Psi}{\partial \theta \partial z}, \tag{14}$$

$$u_z = \frac{\partial \Phi}{\partial z} - a\left(\frac{\partial^2 \Psi}{\partial r^2} + \frac{1}{r}\frac{\partial \Psi}{\partial r} + \frac{1}{r^2}\frac{\partial^2 \Psi}{\partial \theta^2}\right). \tag{15}$$

Turbulence is generated through the pinhole, but it is maximized at the leaking orifice due to edge discontinuities. Since the hole cross-sectional area is very small compared to

the cylinder surface, the force due to turbulent outflow at the orifice surface can be treated as PS. As the PS, the force $f$ due to FRS is expressed as follows

$$f = P_w \delta(x - x_0) e^{-i\omega t}. \tag{16}$$

In Equation (16), $\omega$ is the angular frequency that transmits FRS energy from the PS to the cylinder and resonates with the energy of cylinder materials. As a solution to the delta function, Green's function $g(x; x_0)$ is defined as

$$\nabla^2 g(x; x_0) = \delta(x - x_0). \tag{17}$$

In cylindrical coordinates, Equation (17) is expressed as

$$\nabla^2 g(r, \theta, z; r_0, \theta_0, z_0) = \frac{\delta(r - r_0)\delta(\theta - \theta_0)\delta(z - z_0)}{r}, \tag{18}$$

and the solution of Equation (18) is given as [18]

$$g(r, \theta, z; r_0, \theta_0, z_0) = A_{v1} g_r(r; r_0) g_\theta(\theta; \theta_0) g_z(z; z_0), \tag{19}$$

where $A_{v1}$ is the coupling constant, expressed as the first root of the Bessel function in $g_r$ and the integer $v$ of the aperiodicity in $g_\theta$. To make the position of the PS the new origin, as shown in Figure 1b, the coordinates $(r, \theta, z; r_0, \theta_0, z_0)$ are replaced by $(\xi, \vartheta, \eta; 0)$, defined as

$$\xi_i = x_i - x_{0i}, \ \ \xi_j = x_j - x_{0j} = x_j, \ \ \vartheta = \theta - \theta_0, \ \text{and} \ \eta = z - z_0,$$

where the PS is located on the $x_i$ axis, $r_0 = x_{0i}$. Previously [18], we derived the Green's function responsible for the PS located inside the cylinder. Assuming that the Green's function is periodic ($v = 0$)

$$g(\xi, \vartheta, \eta) = G_{01} J_0(\kappa_z \xi), \tag{20}$$

$$G_{01} = -\frac{1}{2\kappa_z} A_{01} J_0(\kappa_z \xi_0) \begin{cases} e^{-\kappa_z \eta} & (0 \le \eta \le l - z_0) \\ e^{\kappa_z \eta} & (-z_0 \le \eta < 0) \end{cases}, \tag{21}$$

$$A_{01} = \frac{1}{\pi \left[ e^{\kappa_z z_0} + e^{-\kappa_z (l - z_0)} - 2 \right]} \times \frac{2}{\left\{ a^2 [J_{v+1}(\kappa_z a)]^2 - b^2 [J_{v+1}(\kappa_z b)]^2 \right\}}, \tag{22}$$

where $l$ is the length of the cylinder, and $a$ and $b$ are the outer and inner diameters, respectively. (Note: this corrects the erratum in Equation (16) for the $\eta$ range in Ref. [18].) For gas leakage, the Green's function is non-zero on the outer surface of the cylindrical shell because the PS is located on the outer surface of the cylindrical shell. In this study, the value of $\kappa_z$ was determined empirically.

In Equation (20), the value of $\xi$ is the shortest distance between the PS and the point where the detecting sensor is projected onto the equatorial layer containing the PS. However, as shown in Figure 1b, there is no linear distance between the two points across the hollow interior. Since the thickness is much shorter than the diameter of the outer circle, we simply use the value of $\xi$ as the length of the arc around the outer circle,

$$\xi \approx \xi_p = \pi a \theta / 180°, \tag{23}$$

where $\theta$ is angle between the PS and the projected point.

From Equations (16) and (17), the force vector can be rewritten as

$$\begin{aligned} f &= P\nabla^2 g(\xi, \vartheta, \eta) e^{-i\omega t} \\ &= \nabla[\nabla \cdot P g(\xi, \vartheta, \eta)] - \nabla \times [\nabla \times P g(\xi, \vartheta, \eta)] e^{-i\omega t}, \end{aligned} \tag{24}$$

where $\nabla^2 = \frac{\partial^2}{\partial \xi^2} + \frac{1}{\xi}\frac{\partial}{\partial \vartheta} + \frac{1}{\xi^2}\frac{\partial^2}{\partial \vartheta^2} + \frac{\partial^2}{\partial \eta^2}$.

Previously [18], the three potential functions were specified as CFIPs generated by the PS.

$$\Phi = \boldsymbol{\nabla}\cdot(\boldsymbol{P}\phi) = \frac{\partial(\boldsymbol{P}\phi)}{\partial\xi} + \frac{1}{\xi}\frac{\partial(\boldsymbol{P}\phi)}{\partial\vartheta} + \frac{\partial(\boldsymbol{P}\phi)}{\partial\eta}, \tag{25}$$

$$X\hat{e}_\eta = -\boldsymbol{\nabla}\times(\boldsymbol{P}\chi) = -\frac{1}{\xi}\left[\frac{\partial(\xi\boldsymbol{P}\chi)}{\partial\xi} - \frac{\partial(\boldsymbol{P}\chi)}{\partial\vartheta}\right]\hat{e}_\eta, \tag{26}$$

$$\Psi\hat{e}_\eta = -\frac{1}{\xi}\left[\frac{\partial(\xi\boldsymbol{P}\psi)}{\partial\xi} - \frac{\partial(\boldsymbol{P}\psi)}{\partial\vartheta}\right]\hat{e}_\eta, \tag{27}$$

where $\phi$, $\chi$ and $\psi$ are scalar functions. These scalar functions were determined by solving Equation (10) combined with Equations (11) and (25)–(27) as follows

$$\phi(\xi,\vartheta,\eta) = A_m J_m(\alpha\xi)\cos(m\vartheta)e^{-ik_\eta\eta} - \frac{k_p^2}{\rho\omega^2}\frac{G_{01}}{\alpha^2 - \kappa_z^2}J_m(\kappa_z\xi), \tag{28}$$

$$\chi(\xi,\vartheta,\eta) = B_m J_m(\beta\xi)\sin(m\vartheta)e^{-ik_\eta z} - \frac{k_p^2}{\rho\omega^2}\frac{G_{01}}{\beta^2 - k^2}J_m(\kappa_z\xi), \tag{29}$$

$$\psi(\xi,\vartheta,\eta) = C_m J_m(\beta\xi)\cos(m\vartheta)e^{-ik_\eta\eta}, \tag{30}$$

where $\alpha^2 = k_p^2 - k_\eta^2 > 0$, and $\beta^2 = k_s^2 - k_\eta^2 > 0$. In Equations (28)–(30), $A_m$, $B_m$ and $C_m$ are the coupling constants.

As shown in Figure 1a, the CF vectors generated by the FRS act in the axial ($P_z$) and the tangential ($P_\theta$) directions. There is no CF in the radial ($P_r$) direction because the surface boundary is perpendicular to the radial direction. Substituting Equation (28) into Equation (25), the CFIPs for the P wave is given by

$$\Phi_z = P_{\mathrm{w}}\left[(-ik_\eta)A_m J_m(\alpha\xi)\cos(m\vartheta)e^{-ik_\eta\eta} - \frac{k_p^2}{\rho\omega^2}\frac{J_m(\kappa_z\xi)}{\alpha^2 - \kappa_z^2}\frac{\partial G_{01}}{\partial\eta}\right], \tag{31}$$

$$\Phi_\theta = -P_{\mathrm{w}}\left[A_m m\frac{1}{\xi}J_m(\alpha\xi)\sin(m\vartheta)e^{-ik_\eta\eta}\right], \tag{32}$$

similarly, substituting Equation (29) into Equation (26), we obtain the CFIPs for the SH wave as

$$X_z = 0, \tag{33}$$

$$X_\theta = -P_{\mathrm{w}}\left\{B_m m J_m(\beta\xi)\cos(m\vartheta)e^{-ik_\eta\eta} - \frac{k_p^2}{\rho\omega^2}\frac{G_{01}}{\beta^2 - k^2}\left[J_m(\kappa_z\xi) + \xi\frac{\partial J_m(\kappa_z\xi)}{\partial\xi}\right]\right\}. \tag{34}$$

Substituting Equation (30) into Equation (27), the CFIPs for the SV wave is given by

$$\Psi_z = 0, \tag{35}$$

$$\Psi_\theta = -P_{\mathrm{w}}C_m\left[\frac{1}{\xi}J_m(\beta\xi) + \frac{\partial J_m(\beta\xi)}{\partial\xi}\right]\cos(m\vartheta)e^{-ik_\eta\eta}, \tag{36}$$

since all CFIPs for the P, SH, and SV waves at a given CF direction have been fully derived due to pinhole leakage, the displacement components in the $(\xi,\vartheta,\eta)$ coordinates generated by the CFIPs can be obtained using the component form in Equations (13)–(15). Note: in the same way as in Ref. [18], the component of the displacement $d$ due to $P_f$ is expressed as

$$u_{df} = P_f\left(A_{mf}F_{df}^1 + B_{mf}F_{df}^2 + C_{mf}F_{df}^3 + F_{df}^4\right), \tag{37}$$

where $P_f$ is $P_z$ or $P_\theta$.

For $P_z$, the radial components $u_{rz}$,

$$F^1_{rz} = (-ik_\eta) \frac{\partial J_m(\alpha\xi)}{\partial\xi} \cos(m\vartheta)e^{-ik_\eta\eta}, \tag{38}$$

$$F^2_{rz} = F^3_{rz} = 0, \tag{39}$$

$$F^4_{rz} = -\frac{k_p^2}{\rho\omega^2} \frac{1}{\alpha^2 - \kappa_z^2} \frac{\partial J_m(\kappa_z\xi)}{\partial\xi} \frac{\partial G_{01}}{\partial\eta}, \tag{40}$$

the tangential components $u_{\theta z}$,

$$F^1_{\theta z} = (ik_\eta) \frac{m}{\xi} J_m(\alpha\xi)\sin(m\vartheta)e^{-ik_\eta\eta}, \tag{41}$$

$$F^2_{\theta z} = F^3_{\theta z} = F^4_{\theta z} = 0, \tag{42}$$

and the axial component $u_{zz}$,

$$F^1_{zz} = -k_\eta^2 J_m(\alpha\xi)\cos(m\vartheta)e^{-ik_\eta\eta}, \tag{43}$$

$$F^2_{zz} = F^3_{zz} = 0, \tag{44}$$

$$F^4_{zz} = -\frac{k_p^2}{\rho\omega^2} \frac{1}{\alpha^2 - \kappa_z^2} J_m(\kappa_z\xi) \frac{\partial^2 G_{01}}{\partial\eta^2}, \tag{45}$$

(Note: this corrects the erratum in Equation (62) for $F^4_{zz}$ in Ref. [18].)

For $P_\theta$, the radial components $u_{r\theta}$,

$$F^1_{r\theta} = m\left[\frac{1}{\xi^2} J_m(\alpha\xi) - \frac{1}{\xi}\frac{\partial J_m(\alpha\xi)}{\partial\xi}\right]\sin(m\vartheta)e^{-ik_\eta\eta}, \tag{46}$$

$$F^2_{r\theta} = -m\left[\frac{1}{\xi^2} J_m(\beta\xi) + \frac{1}{\xi}\frac{\partial J_m(\beta\xi)}{\partial\xi}\right]\cos(m\vartheta)e^{-ik_\eta\eta}, \tag{47}$$

$$F^3_{r\theta} = (ik_\eta)a\left[\frac{1}{\xi^2} J_m(\beta\xi) - \frac{1}{\xi}\frac{\partial J_m(\beta\xi)}{\partial\xi} - \frac{\partial^2 J_m(\beta\xi)}{\partial\xi^2}\right]\cos(m\vartheta)e^{-ik_\eta\eta}, \tag{48}$$

$$F^4_{r\theta} = 0, \tag{49}$$

the tangential components $u_{\theta\theta}$,

$$F^1_{\theta\theta} = -\frac{m^2}{\xi^2} J_m(\alpha\xi)\cos(m\vartheta)e^{-ik_\eta\eta}, \tag{50}$$

$$F^2_{\theta\theta} = \left[-\frac{1}{\xi^2} J_m(\beta\xi) + \frac{1}{\xi}\frac{\partial J_m(\beta\xi)}{\partial\xi} + \frac{\partial^2 J_m(\beta\xi)}{\partial\xi^2}\right]\sin(m\vartheta)e^{-ik_\eta\eta}, \tag{51}$$

$$F^3_{\theta\theta} = -\frac{a}{\xi}(ik_\eta)m\left[\frac{1}{\xi} J_m(\beta\xi) + \frac{\partial J_m(\beta\xi)}{\partial\xi}\right]\sin(m\vartheta)e^{-ik_\eta\eta}, \tag{52}$$

$$F^4_{\theta\theta} = \frac{k_p^2}{\rho\omega^2} \frac{1}{\beta^2 - k^2} G_{01}\left\{\frac{1}{\xi^2}\left[J_m(\kappa_z\xi) + \frac{\partial J_m(\kappa_z\xi)}{\partial\xi}\right] - \frac{1}{\xi}\left[2\frac{\partial J_m(\kappa_z\xi)}{\partial\xi} + \frac{\partial^2 J_m(\kappa_z\xi)}{\partial^2\xi}\right]\right\}, \tag{53}$$

and the axial component $u_{z\theta}$,

$$F_{z\theta}^1 = \frac{ik_\eta}{\zeta} m J_m(\alpha\zeta)\sin(m\vartheta)e^{-ik_\eta\eta}, \tag{54}$$

$$F_{z\theta}^2 = 0, \tag{55}$$

$$F_{z\theta}^3 = a\left[\left(\frac{1-m^2}{\zeta^3}\right)J_m(\beta\zeta) - \left(\frac{1+m^2}{\zeta^2}\right)\frac{\partial J_m(\beta\zeta)}{\partial\zeta} + \frac{2}{\zeta}\frac{\partial^2 J_m(\beta\zeta)}{\partial\zeta^2} + \frac{\partial^3 J_m(\beta\zeta)}{\partial\zeta^3}\right]\cos(m\vartheta)e^{-ik_\eta\eta}, \tag{56}$$

$$F_{z\theta}^4 = 0, \tag{57}$$

the remaining task is to evaluate the coupling constants $A_m$, $B_m$ and $C_m$. Let us apply boundary conditions to the outer surface in the same way as in Ref. [18]. There is no stress on the outer surface of the cylinder because the effect of the atmosphere pressure on the displacement field is negligible

$$\sigma_{rr} = \sigma_{r\theta} = \sigma_{rz} = 0. \tag{58}$$

Introducing Equation (58) to the stress–strain displacement relations for the circular cylindrical shell studied in this work yields a system of three linear algebraic equations, given as follows:

$$\begin{bmatrix} a_{11f} & a_{12f} & a_{13f} \\ a_{21f} & a_{22f} & a_{23f} \\ a_{31f} & a_{32f} & a_{33f} \end{bmatrix}\begin{bmatrix} A_{mf} \\ B_{mf} \\ C_{mf} \end{bmatrix} = \begin{bmatrix} b_{1f} \\ b_{2f} \\ b_{3f} \end{bmatrix}. \tag{59}$$

Elements $a_{ijf}$ and $b_{if}$ of the matrix in Equation (59) are given in Appendix A. For the $P_z$ CF, Equation (59) becomes very simple because all elements except $a_{1z}$, $b_{1z}$ and $b_{3z}$ are zero. We obtain

$$B_{mz} = 0, \tag{60}$$

$$A_{mz} = \frac{b_{1z}}{a_{11z}} = \frac{b_{3z}}{a_{31}}. \tag{61}$$

The values of $k_\eta$ at a given location of the PS can be obtained by solving the roots of the function

$$f(k_\eta) = \frac{b_{1z}}{a_{11z}} - \frac{b_{3z}}{a_{31z}} = 0. \tag{62}$$

To evaluate the displacements at position $(\xi, \eta)$ on the outer surface, the arrival times ($\tau$s) of the AE signal generated by the PS force must be introduced into Equation (37). The arrival times of the P and S waves propagating with velocities $c_P$ and $c_S$ are given as

$$\tau_P = \frac{\sqrt{\xi^2 + \eta^2}}{c_P}, \ \tau_P = \frac{\sqrt{\xi^2 + \eta^2}}{c_S}, \tag{63}$$

respectively, where $\xi$ is given by Equation (23). Finally, the displacement fields generated by the gas leakage can be summarized as follows

$$u_{df} = P_f\left[(t - \tau_P)\left(A_{mf}F_{df}^1 + F_{df}^4\delta_{dr}\right) + (t - \tau_S)\left(B_{mf}F_{df}^2 + C_{mf}F_{df}^3 + F_{df}^4\delta_{d\theta}\right)\right] \times e^{-i\omega t}, \tag{64}$$

where $d = r$, $\theta$ and $z$, and $f = z$ and $\theta$. Based on the wave characteristics, Equation (64) can be divided into the P, SH and SV waves as

$$u_{df}^P = P_f(t - \tau_P)\left(A_{mf}F_{df}^1 + F_{df}^4\delta_{dr}\right)e^{-i\omega t}, \tag{65}$$

$$u_{df}^{SH} = P_f(t - \tau_S)\left(B_{mf}F_{df}^2 + F_{df}^4 \delta_{d\theta}\right)e^{-i\omega t}, \tag{66}$$

$$u_{df}^{SV} = P_f(t - \tau_S)\left(C_{mf}F_{df}^3\right)e^{-i\omega t}. \tag{67}$$

### 3. Experiment

The cylinder shell used in this study is a seamless $N_2$ gas cylinder (40 L, Mn steel) manufactured according to KGS AC 212 specification [25]. The length of the shell is 1185 mm, of which the main part with the same cross-sectional area is 1013 mm, and the outer diameter and the thickness are 232 mm and 4.8 mm, respectively (Figure 2). The leak source (LS) was created by inserting hexagonal head cap screw with a hole into a bore-hole in the main segment wall, located at 0.368 m from the front of the main part (the distance from the LS to the end plate is 0.817 m). The diameters of the holes in the screws comprised 0.2, 0.3, 0.50, 0.80, and 1.2 mm. The pressure in the cylinder was maintained at a given value by two gas regulators connected to a supply high-pressure cylinder. The pressure in the cylinder was read and recorded by a digital pressure gauge (PDR500, PDK, Daejeon, Republic of Korea).

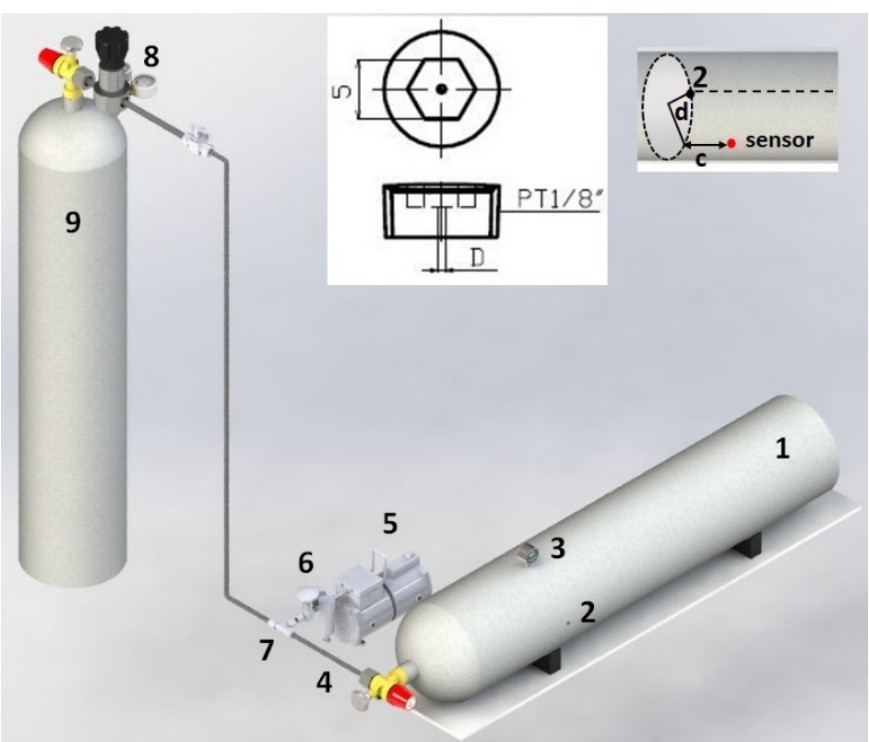

**Figure 2.** AE experimental schematic diagram: 1, test cylinder; 2, LS; 3, pressure gauge; 4, cylinder valve; 5, vacuum pump; 6, needle valve; 7, on–off valve; 8, pressure regulator system; 9, $N_2$ gas storage (insertion: left, details of hexagonal head cap screw with a circular hole as LS with 5 mm width and D mm hole; right, position of a sensor represented by $\eta$ = c mm and $\vartheta$ = d°).

To detect AE signals, several broadband AE sensors (IDK-AES-H150, IDK, Daejeon, Republic of Korea) with a resonant frequency of 150 kHz were used. These sensors can detect any AE signals in a range of 0–500 kHz. AE acquisition and analysis was performed using an IDK-AET-H150/E08 system, which includes a pre-amplifier, an 8-channel DAQ board, and computer data storage (post-processors). The IDK System generates acoustic data through sequential amplification and FFT of detected electric signals, and runs the embedded AE Studio Software (v1.6, IDK). No filters were used, and the signal threshold was 40 dB. AE hits were captured over its full acoustic range during 300 s. Each hit

represents an AE signal, allowed by a timer controller with 1.0 ms of maximum hit duration in the DAQ board.

## 4. Results and Discussion

### 4.1. AE Properties

The AE characteristics generated by the gas leak were investigated as a functions of gas pressure ($p_0$), axial distance ($\eta$), and angle ($\vartheta$) to the LS position (for convenience, the experimental test is presented as DaPb$\eta$c$\vartheta$d, where a, b, c and d denote the diameter of LS (mm), internal pressure (bar), axial distance from the PS (cm), and the tangential angle (degree), respectively). The cylinder internal pressure was kept at a given b bar. The AE signals were measured as waveforms. From observed signals, we evaluated several useful AE parameters such as count, maximum amplitude, duration, signal RMS, rise time, signal energy, peak frequency, etc. Figure 3a,c show two typical AE signal waveforms captured at D0.2P2$\eta$1$\vartheta$0 and D1.2P4$\eta$1$\vartheta$0, respectively. To reveal the resonant frequency of the AE signal transmitted through the surface of the cylinder, the time-domain spectra were converted to the frequency-domain spectra by applying FFT. After applying the Savitzky–Golay smoothing method, the amplitude FFT spectra were resolved by fitting with a Gaussian function. As shown in Figure 3b, the FFT spectrum of the AE signal of D0.2P2$\eta$1$\vartheta$0 consists of a strong peak at 173.4 kHz with several weak peaks in the 30–340 kHz region. Among them, the shortest and the longest frequencies appeared at 70.2 kHz and 284.5 kHz, respectively. The 173.4 kHz and the 145.6 kHz bands have weights of 45% and 18%, respectively, while the remaining bands have weights of less than 10%. In contrast, for the AE of D1.2P4$\eta$1$\vartheta$0, the main band at 108.1 kHz appeared with a weight of 43%, and the 171.2 kHz band appeared as the second peak with 19 wt.% (Figure 3d). It is important to note that the 108.1 kHz band appeared as an additional band for the AE signal from D0.2P2$\eta$1$\vartheta$0.

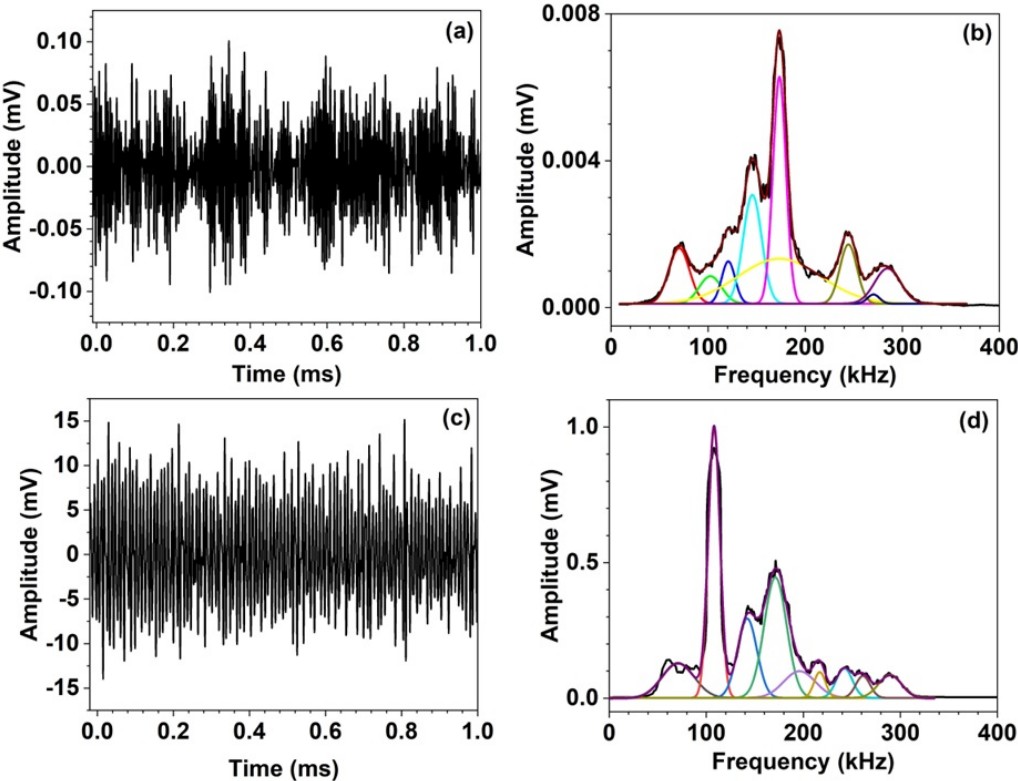

**Figure 3.** Typical AE signals (**a**,**c**) and their resolved FFT spectra (**c**,**d**) observed by D0.2P2$\eta$1$\vartheta$0 (**a**,**b**) and D1.2P4$\eta$1$\vartheta$0 (**c**,**d**), respectively.

The distribution of the mean frequency for the AE signals from D0.2P2η1ϑ0 and D1.2P4η1ϑ0 is illustrated in Figure 4. Except for the signals from D1.2P4η1ϑ0, all the acquired AE signals show the peak frequencies in the range of 170–175 kHz. For D1.2 with $\vartheta \neq 0$, the peak frequencies in the range of 60–70 kHz were seen with weights less than 40%.

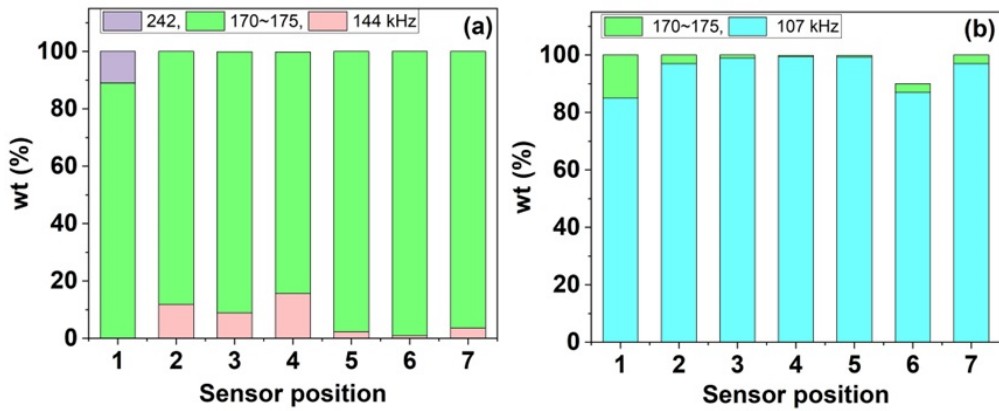

**Figure 4.** Two types of peak-frequency distributions, observed in (**a**) most tests, and (**b**) only D1.2P4.

### 4.2. Angular and Axial Dependence

The properties of AE signals were investigated as a function of tangential angle at $\eta = 1$ cm. Seven AE sensors were mounted at a 30° interval from $\vartheta = 0°$ to 180°. The mean value of the amplitude was calculated from over 70 k hits. Figure 5a shows the mean amplitudes with σ values generated from D0.2P4η1ϑ0 and D1.2P4η1ϑ0. For D0.2P4η1ϑ0, the amplitude of the AE signals is 0.58 (σ = 0.087) mV at $\vartheta = 0°$. As the angle increased, the amplitude decreased, reaching a minimum between 60° and 90°. Above this angle, the amplitude increased, reaching a maximum at ~120° and a second minimum at ~150°. At $p_0 = 2$ bar, similar angular-dependence features were observed for D0.2–D0.5, while different features showing the minimum at ~120° were found for D0.5 and D1.2 (Figure A2).

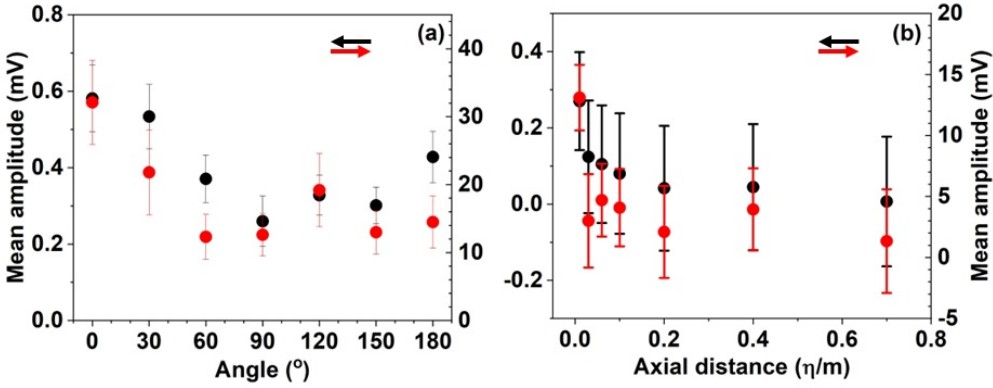

**Figure 5.** Mean amplitudes (circles) and σs (bars) of AE signals generated from: (**a**) D0.2P4η1 (black) and D1.2P4η1 (red) at $\vartheta = 0 - 180$, and (**b**) D0.2P4ϑ0 (black) and D1.2P4ϑ0 (red) at $\eta = 1 - 70$ cm.

The axial dependence of the AE amplitude was also investigated using D0.2–D1.2 LSs at $\vartheta = 0°$. Seven AE sensors were mounted at downstream locations aligned with the LS along the z axis on the cylinder surface. Figure 5b shows the observed results for D0.2P4ϑ0 and D1.2P4ϑ0. For D0.2P4ϑ0 LS, the baseline subtracted (BS) amplitude was 0.27 mV at $\eta = 1$ cm. As η increased to 3 cm, the BS amplitude decreased sharply to 0.124 mV. Above $\eta = 3$ cm, the BS amplitude slowly decreased with increasing η. It can be seen that the mean amplitudes of the AE signals generated from D0.2P4ϑ0 LS clearly exhibit exponential-decay

characteristics. For D1.2P4$\vartheta$0 LS, some of the data fluctuated to some extent; however, it is satisfied the exponential-decay characteristics within the standard deviation.

### 4.3. Verification of CFIP Model

A stainless steel cylindrical structure ($\rho = 7.80 \times 10^3$ kg/m$^3$, $c_p = 5.98$ km/s, $c_s = 3.30$ km/s) was used for the simulation.

#### 4.3.1. Determination of $\kappa_z$

Before simulating the displacement field, the relation between $k_\eta$ and $\kappa_z$ was determined from Equation (61). As shown in Figure A3, as the value of $\kappa_z$ increased, the value of $k_\eta$ gradually decreased until $\kappa_z = 15$, while above $\kappa_z = 15$, the value of $k_\eta$ is almost independent of the $\kappa_z$ value. The angular-dependence of the simulated displacement field is very effective in determining the $\kappa_z$ value. The angular dependence of the observed AE amplitudes appears to be related to the Bessel function $J_0(\kappa_z\xi)$, which is involved in the Green's function in Equation (20). The Bessel function goes to zero at a certain value of $\xi$, depending on the value of $\kappa_z$. As shown in Figure A4, no minimum appears until $\kappa_z = 2$. Above this value, a single minimum appears at $\vartheta = 176°$ for $\kappa_z = 3$, $\vartheta = 141°$ for $\kappa_z = 5$, and $\vartheta = 87°$ for $\kappa_z = 8$. Above $\kappa_z = 8$, a well of multiple minima appears: for $\kappa_z = 9$, there are two minima position at $\vartheta = 78°$ and $164°$. In the simulation, we set $\kappa_z = 9$, and set FRS = 91 N/m$^2$ for $p_0 = 2$ bar and 250 N/m$^2$ for $p_0 = 4$ bar.

#### 4.3.2. Multi-Frequency AE

The observed angular dependency led us to conclude that the displacement fields generated by a gas leakage in the cylindrical geometry is $2\pi$-periodic: $m = 0$. Therefore, Equation (64) is rewritten as

$$
\begin{aligned}
u_{df} = P_f \Big[ (t - \tau_P)\Big( A_{0f}F_{df}^1 + F_{df}^4\delta_{dr} \Big) \\
+ (t - \tau_S)\Big( B_{0f}F_{df}^2 + C_{0f}F_{df}^3 + F_{df}^4\delta_{d\theta} \Big) \Big] e^{-i\omega t},
\end{aligned}
\tag{68}
$$

The displacement field, expressed by Equation (68), indicates that the AE wave is active if $F_{df}^4$ is nonzero. From Equations (40), (42), (45), (49), (53) and (57), it can be found that $F_{rz}^4$, $F_{zz}^4$ and $F_{\theta\theta}^4$ are nonzero, but $F_{\theta z}^4$, $F_{r\theta}^4$ and $F_{z\theta}^4$ are zero. Considering that the nonzero $F_{df}^4$ terms are proportional to $1/\omega^2$, Equation (68) can be rewritten for multi-frequency waves as

$$
u_{df}(\omega_1, \cdots, \omega_n) = \sum_{i=1}^{n} W_i u_{df}(\omega_i),
\tag{69}
$$

$$
W_i = \frac{\omega_i^2}{\sum_i^n \omega_i^2}.
\tag{70}
$$

Equation (70) obtains contributions with equal weights at the configured frequencies. Therefore, the observed weight of the $i$ component $(w_{\text{emp},i})$ is compensated to Equation (70). The final formula of the AE wave is

$$
u_{df}(\omega_1, \cdots, \omega_n) = \sum_{i=1}^{n} w_{\text{emp},i} \frac{\omega_i^2}{\sum_i^n \omega_i^2} u_{df}(\omega_i),
\tag{71}
$$

where $u_{rz}$, $u_{zz}$ and $u_{\theta\theta}$ are active and the others are inactive. Among these active displacements, $u_{zz}$ is dominant (hereafter, referred to as $u_z$).

#### 4.3.3. Simulation

The AE wave shown in Figure 3a was chosen as the model for the simulation. The frequency components and their observed weights are listed in Table 1. In the simulation, these data were introduced into Equation (71). Figure 6a shows the simulated $u_z$ displacement and corresponding FFT spectrum for D0.2P4$\eta$1$\vartheta$0.001 (FRS = 250 N/m$^2$) using the

eight frequencies listed in Table 1. Since $g(\xi, \vartheta, \eta)$ diverges to infinity at $\vartheta = 0$, we set the minimum angle to 0.001° in the simulation. As shown in Figure 6b, the FFT spectrum of the simulated AE wave is in good agreement with the observed spectrum (Table 1).

**Table 1.** Frequency components and their weights used in the simulation.

| $\nu_i$/kHz | 70.2 | 102.3 | 120.8 | 145.6 | 173.4 | 244.5 | 270.3 | 284.5 |
|---|---|---|---|---|---|---|---|---|
| $w_{\text{emp},i}$ | 0.091 | 0.045 | 0.069 | 0.18 | 0.45 | 0.097 | 0.015 | 0.058 |

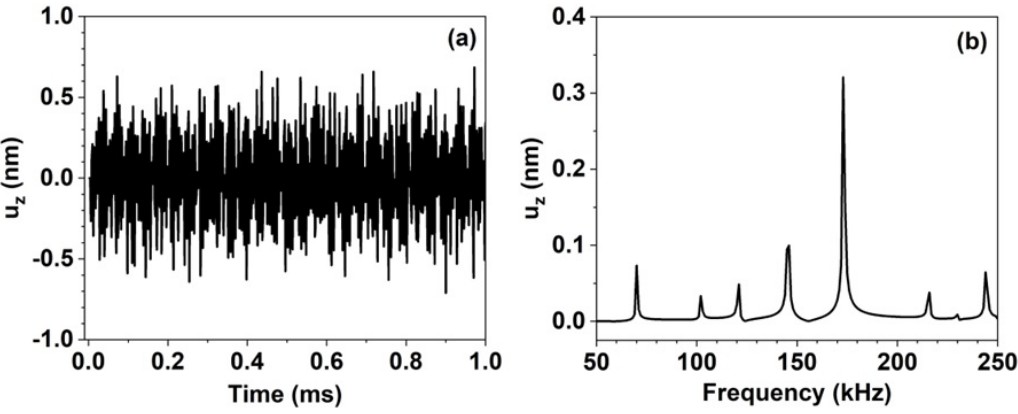

**Figure 6.** (**a**) Displacement field, $u_z$, simulated with FRS = 250 N/m² and $\kappa_z = 9$, and (**b**) corresponding FFT spectrum.

Here, we demonstrated the validity of our approach by evaluating the multi-frequency $u_z$ as a function of $\vartheta$ for angular dependence and a function of $\eta$ for axial dependence. The maximum value was taken from the simulated $u_z$ displacements. For the angular dependency, both experimental values and calculated values (after baseline correction) were converted to relative values for a value of $\vartheta = 120°$. A baseline correction to the experimental values was necessary to match the two values, because the calculated minimum was nearly zero (Figure A4). As shown in Figure 7, the results for the angular dependency show good agreement between the observed and simulated values.

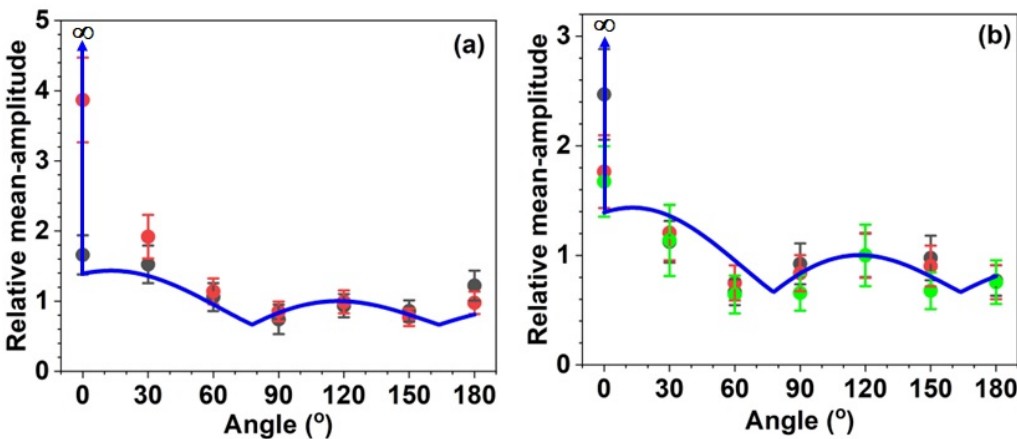

**Figure 7.** Relative values of observed mean amplitudes (solid circles) and σs (bars) of simulated maximum $u_z$ displacements (lines): (**a**) D0.2P4η1 (black) and D0.3P4η1 (red), (**b**) D0.5P4η1 (black), D0.8P4η1 (red) and D1.2P4η1 (green). In the simulation, $\kappa_z = 9$ and FRS = 250 N/m².

For the axial dependence, the simulation was also performed at $\vartheta = 0.001$ instead of $\vartheta = 0$. Simulated values given in meter were converted to those given in mV by multiplying the appropriate factor. The results for D0.2 and D1.2 are shown in Figure 8, and the others

are shown in Figure A5. When η approaches zero, there is a sudden increase in the observed AE amplitude. The experimental results show that the AE signal increases abnormally when approaching the leakage point.

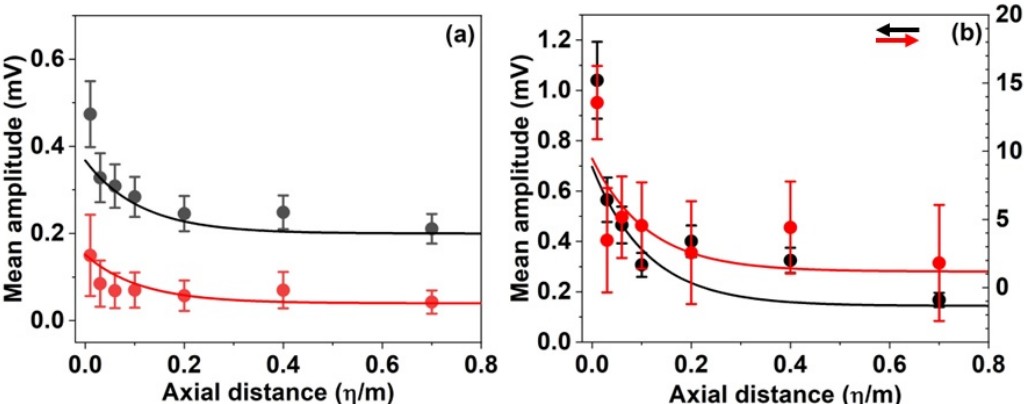

**Figure 8.** Observed mean amplitudes (solid circles) and σs (bars) of simulated maximum $u_z$ displacements (lines): (**a**) D0.2P2 $\vartheta$0 (black) and D0.2P4$\vartheta$0 (red), (**b**) D1.2P2$\vartheta$0 (black) and D1.2P4$\vartheta$0 (red). In the simulation, $\kappa_z = 9$, and FRS = 250 N/m$^2$ for P4 and 91 N/m$^2$ for P2. The simulated values were multiplied by $7.32 \times 10^7$ mV/m for D0.2P2 and D0.2P4, and $5.69 \times 10^8$ mV/m for D1.2P2 and $3.60 \times 10^9$ mV/m for D1.2P4.

## 5. Conclusions

In this paper, we provided a mathematical model for AE due to leakage in a cylindrical vessel. As in an internal crack, the leakage generates CFIPs for P, SH, and SV waves. The fluctuating Reynolds stress caused by pinhole leakage in a gas reservoir was introduced as a point source into the NL equation, and the solutions for radial, tangential, and axial displacements were obtained. In addition, the angular and axial dependencies of the AE signals generated by pinhole leakage sources of different diameters were investigated to reveal their characteristic features. The main advantage of this study is that it provides an accurate solution for the AE characteristics caused by leakage in cylindrical geometries. In the near future, this mathematical model will be used to locate leakage sources with more experimental data.

**Author Contributions:** Conceptualization, methodology and investigation, K.B.K., J.-H.K., J.-E.J., C.-I.K., B.K.K. and J.-G.K.; soft-ware, validation, and formal analysis, K.B.K., H.-J.K. and C.-I.K.; resources and data curation, K.B.K., J.-H.K., J.-E.J. and J.-G.K.; writing—original draft preparation, K.B.K. and J.-G.K.; writing—review and editing, visualization, and supervision, J.-G.K.; project administration and funding acquisition, B.K.K. All authors have read and agreed to the published version of the manuscript.

**Funding:** This research was funded by the Korea Institute of Energy Technology Evaluation and Planning (KETEP) and the Ministry of Trade, Industry & Energy (MOTIE) of the Republic of Korea (NO. 2022303004020A).

**Institutional Review Board Statement:** Not applicable.

**Informed Consent Statement:** Not applicable.

**Data Availability Statement:** Not applicable.

**Conflicts of Interest:** The authors declare no conflict of interest.

**Appendix A**

For a transversely isotropic cylinder, the three stresses are given as

$$\sigma_{rrf} = c_{11}\left(\frac{\partial u_{rf}}{\partial \xi}\right) + c_{12}\left(\frac{u_{rf}}{\xi} + \frac{1}{\xi}\frac{\partial u_{\theta f}}{\partial \vartheta}\right) + c_{13}\left(\frac{\partial u_{zf}}{\partial \eta}\right)$$
$$= P_f\left(a_{11f}A_{mf} + a_{12f}B_{mf} + a_{13f}C_{mf} + b_{1f}\right)e^{-i\omega t}, \tag{A1}$$

$$\sigma_{r\theta f} = \frac{(c_{11}-c_{12})}{2}\left(\frac{\partial u_{\theta f}}{\partial \xi} - \frac{u_{\theta f}}{\xi} + \frac{1}{\xi}\frac{\partial u_{rf}}{\partial \vartheta}\right)$$
$$= P_f\left(a_{21f}A_{mf} + a_{22f}B_{mf} + a_{23f}C_{mf} + b_{2f}\right)e^{-i\omega t}, \tag{A2}$$

$$\sigma_{rzf} = c_{44}\left(\frac{\partial u_{zf}}{\partial \xi} + \frac{\partial u_{rf}}{\partial \eta}\right)$$
$$= P_f\left(a_{31f}A_{mf} + a_{32f}B_{mf} + a_{33f}C_{mf} + b_{3f}\right)e^{-i\omega t}, \tag{A3}$$

where $f = z$ or $\theta$. Substituting Equation (32) into Equations (A1)–(A3) gives the elements $a_{ijf}$ and $b_{if}$ ($i, j = 1$–3)

$$a_{1jf} = c_{11}\frac{\partial F_{rf}^j}{\partial \xi} + c_{12}\frac{F_{rf}^j}{\xi} + c_{12}\frac{1}{\xi}\frac{\partial F_{\theta f}^j}{\partial \vartheta} + c_{13}\left(\frac{\partial F_{zf}^j}{\partial \eta}\right), \tag{A4}$$

$$a_{2jf} = \frac{(c_{11}-c_{12})}{2}\left(\frac{\partial F_{\theta f}^j}{\partial \xi} - \frac{F_{\theta f}^j}{\xi} + \frac{1}{\xi}\frac{\partial F_{rf}^j}{\partial \vartheta}\right), \tag{A5}$$

$$a_{3jz} = c_{44}\left(\frac{\partial F_{zz}^j}{\partial \xi} + \frac{\partial F_{rz}^j}{\partial \eta}\right), \tag{A6}$$

$$b_{1f} = c_{11}\frac{\partial F_{rf}^4}{\partial \xi} + c_{12}\frac{F_{rf}^4}{\xi} + c_{13}\frac{\partial F_{zf}^4}{\partial \eta}, \tag{A7}$$

$$b_{2f} = \frac{(c_{11}-c_{12})}{2}\left(\frac{\partial F_{\theta f}^4}{\partial \xi} - \frac{F_{\theta f}^4}{\xi} + \frac{1}{\xi}\frac{\partial F_{rf}^4}{\partial \vartheta}\right), \tag{A8}$$

$$b_{3f} = c_{44}\left(\frac{\partial F_{zf}^4}{\partial \xi} + \frac{\partial F_{rf}^4}{\partial \eta}\right). \tag{A9}$$

For the $P_z$ CF,

$$a_{11z} = (ik_\eta)\left[\left(c_{12}\frac{m^2}{\xi^2} + c_{13}k_\eta^2\right)J_m(\alpha\xi) - c_{12}\frac{1}{\xi}\frac{\partial J_m(\alpha\xi)}{\partial \xi} - c_{11}\frac{\partial^2 J_m(\alpha\xi)}{\partial \xi^2}\right]$$
$$\times \cos(m\vartheta)e^{-ik_\eta\eta}, \tag{A10}$$

(Note: this corrects the erratum in Equation (77) for $a_{11z}$ in Ref. [18].)

$$a_{12z} = a_{13z} = 0, \tag{A11}$$

$$a_{21z} = \frac{(c_{11}-c_{12})}{2}(ik_\eta)\left[\frac{-2m}{\xi^2}J_m(\alpha\xi) + \frac{2m}{\xi}\frac{\partial J_m(\alpha\xi)}{\partial \xi}\right]\sin(m\vartheta)e^{-ik_\eta\eta}, \tag{A12}$$

$$a_{22z} = a_{23z} = 0, \tag{A13}$$

$$a_{31z} = -2k_\eta^2 c_{44}\frac{\partial J_m(\alpha\xi)}{\partial \xi}\cos(m\vartheta)e^{-ik_\eta\eta}, \tag{A14}$$

$$a_{32z} = a_{33z} = 0, \tag{A15}$$

$$b_{1z} = -\left(\frac{k_p^2}{\rho\omega^2}\right)\frac{1}{\alpha^2 - \kappa_z^2}\left[c_{11}\frac{\partial G_{01}}{\partial\eta}\frac{\partial^2 J_m(\kappa_z\xi)}{\partial\xi^2} + c_{12}\frac{1}{\xi}\frac{\partial G_{01}}{\partial\eta}\frac{\partial J_m(\kappa_z\xi)}{\partial\xi}\right.$$
$$\left. + c_{13}\frac{\partial^3 G_{01}}{\partial\eta^3}J_m(\kappa_z\xi)\right], \tag{A16}$$

$$b_{2z} = 0, \tag{A17}$$

$$b_{3z} = -c_{44}\left(\frac{k_p^2}{\rho\omega^2}\right)\frac{2}{\alpha^2 - \kappa_z^2}\frac{\partial^2 G_1}{\partial\eta^2}\frac{\partial J_m(\kappa_z\xi)}{\partial\xi}. \tag{A18}$$

For the $P_\theta$ CF,

$$a_{11\theta} = m\left[\left(\frac{-2}{\xi^3}c_{11} + \frac{1+m^2\xi}{\xi^3}c_{12} + c_{13}k_\eta^2\right)J_m(\alpha\xi) + \left(\frac{2}{\xi^2}c_{11} - \frac{1}{\xi^2}c_{12}\right)\frac{\partial J_m(\alpha\xi)}{\partial\xi}\right.$$
$$\left. - \frac{1}{\xi}c_{11}\frac{\partial^2 J_m(\alpha\xi)}{\partial\xi^2}\right], \tag{A19}$$

$$a_{12\theta} = m\left[\frac{1}{\xi^2}(2c_{11} - c_{12})J_m(\beta\xi) - \frac{1}{\xi}(c_{11} - c_{12})\frac{\partial J_m(\beta\xi)}{\partial\xi}\right]\sin(m\vartheta)e^{-ik_\eta\eta}, \tag{A20}$$

$$a_{13\theta} = (ik_\eta)a\left\{\frac{1}{\xi^3}\left[-2c_{11} + c_{12}(1-m^2) - c_{13}(1-m^2)\right]J_m(\beta\xi)\right.$$
$$+ \frac{1}{\xi^2}\left[2c_{11} - c_{12}(1-m^2) + c_{13}(1+m^2)\right]\frac{\partial J_m(\beta\xi)}{\partial\xi} - (c_{11} + c_{12} + 2c_{13})\frac{1}{\xi}\frac{\partial^2 J_m(\beta\xi)}{\partial\xi^2}$$
$$\left. - (c_{11} + c_{13})\frac{\partial^3 J_m(\beta\xi)}{\partial\xi^3}\right\}\cos(m\vartheta)e^{-ik_\eta\eta}, \tag{A21}$$

$$a_{21\theta} = (c_{11} - c_{12})m^2\left[\frac{2}{\xi^3}J_m(\alpha\xi) + \frac{1}{\xi^2}\frac{\partial J_m(\alpha\xi)}{\partial\xi} - \frac{1}{\xi^2}\frac{\partial^2 J_m(\alpha\xi)}{\partial\xi^2}\right]\cos(m\vartheta)e^{-ik_\eta\eta}, \tag{A22}$$

$$a_{22\theta} = \frac{(c_{11} - c_{12})}{2}m\left[\frac{m^2}{\xi^2}J_m(\beta\xi) - \frac{1}{\xi}\frac{\partial J_m(\beta\xi)}{\partial\xi} + \frac{\partial^2 J_m(\beta\xi)}{\partial\xi^2}\right]\cos(m\vartheta)e^{-ik_\eta\eta}, \tag{A23}$$

$$a_{23\theta} = \frac{(c_{11} - c_{12})}{2}(ik_\eta)\frac{ma}{\xi}\left[\frac{3}{\xi}J_m(\beta\xi) - \frac{1}{\xi}\frac{\partial J_m(\beta\xi)}{\partial\xi} - 2\frac{\partial^2 J_m(\beta\xi)}{\partial\xi^2}\right]\sin(m\vartheta)e^{-ik_\eta\eta}, \tag{A24}$$

$$a_{31\theta} = -2c_{44}(ik_\eta)\frac{m}{\xi}\left[\frac{1}{\xi}J_m(\alpha\xi) - \frac{\partial J_m(\alpha\xi)}{\partial\xi}\right]\sin(m\vartheta)e^{-ik_\eta\eta}, \tag{A25}$$

$$a_{32\theta} = (ik_\eta)m\left[\frac{1}{\xi^2}J_m(\beta\xi) + \frac{1}{\xi}\frac{\partial J_m(\beta\xi)}{\partial\xi}\right]\cos(m\vartheta)e^{-ik_\eta\eta}, \tag{A26}$$

$$a_{33\theta} = ac_{44}\left\{\left[\frac{3(m^2-1)}{\xi^4} - \frac{k_\eta^2}{\xi^2}\right]J_m(\beta\xi) + \left[\left(\frac{3+m^2}{\xi^3}\right) - \frac{k_\eta^2}{\xi}\right]\frac{\partial J_m(\beta\xi)}{\partial\xi}\right.$$
$$\left. - \left[\left(\frac{3+m^2}{\xi^2}\right) + k_\eta^2\right]\frac{\partial^2 J_m(\beta\xi)}{\partial\xi^2} + \frac{2}{\xi}\frac{\partial^3 J_m(\beta\xi)}{\partial\xi^3} + \frac{\partial^4 J_m(\beta\xi)}{\partial\xi^4}\right\}\cos(m\vartheta)e^{-ik_\eta\eta}, \tag{A27}$$

$$b_{1\theta} = -\left(\frac{k_p^2}{\rho\omega^2}\right)\frac{1}{\alpha^2 - \kappa_z^2}\left[c_{12}\frac{1}{\xi}\frac{\partial^2 G_{01}}{\partial\eta^2}\frac{\partial J_m(\kappa_z\xi)}{\partial\xi} - c_{13}\frac{\partial^3 G_{01}}{\partial\eta^3}J_m(\kappa_z\xi)\right], \tag{A28}$$

$$b_{2\theta} = 0, \tag{A29}$$

$$b_{3\theta} = 0. \tag{A30}$$

## Appendix B

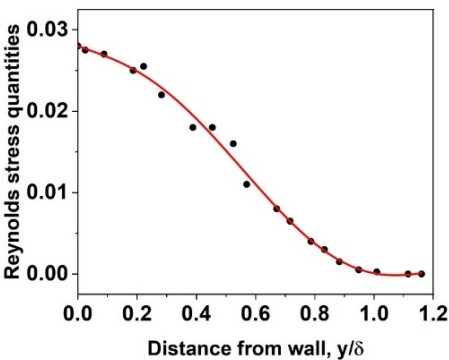

**Figure A1.** Plot of Reynold stress quantity (REQ) vs. distance from wall (y/δ) [20].

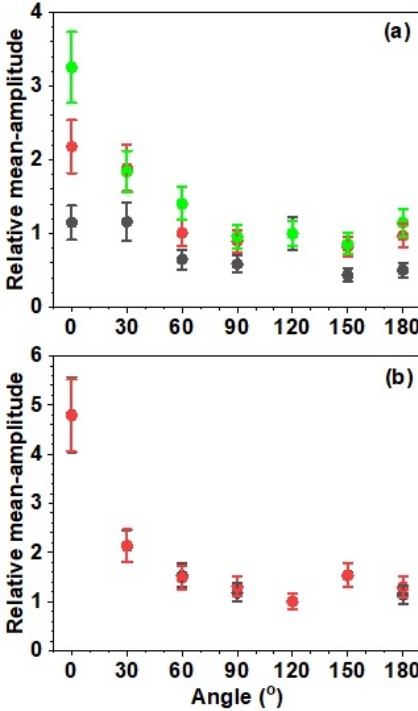

**Figure A2.** Relative mean amplitudes (circles) and σs (bars) of AE signals generated from (**a**) D0.2P2η1 (black), D0.3P2η1 (red) and D0.5P2η1 (green), and (**b**) D0.8P2η1 (black) and D1.2P2η1 (red).

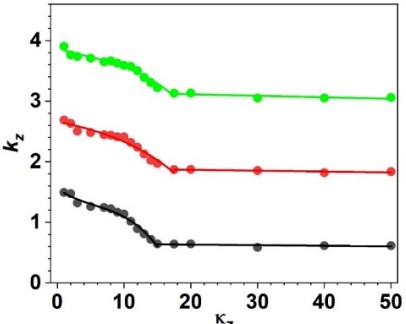

**Figure A3.** First three roots (first: black, second: red, third: green solid circles) of Equation (66) evaluated at some $\kappa_z$ values and polynomial fitting (sold lines). For the first roots, $k_\eta = 1.555 - 0.076\kappa_z + 7.09 \times 10^{-2}\kappa_z^2 - 4.20 \times 10^{-3}\kappa_z^3$ when $\kappa_z \leq 15$, and $k_\eta = 0.652 - 9.95 \times 10^{-3}\,\kappa_z$ when $\kappa_z > 15$.

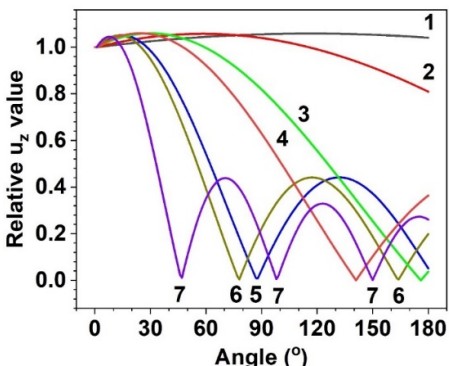

**Figure A4.** Angular dependence of $u_z$ on $\kappa_z$ ($\kappa_z =$: 1; 1, 2; 2, 3: 2, 4; 5, 5; 8, 6; 9, 7; 15).

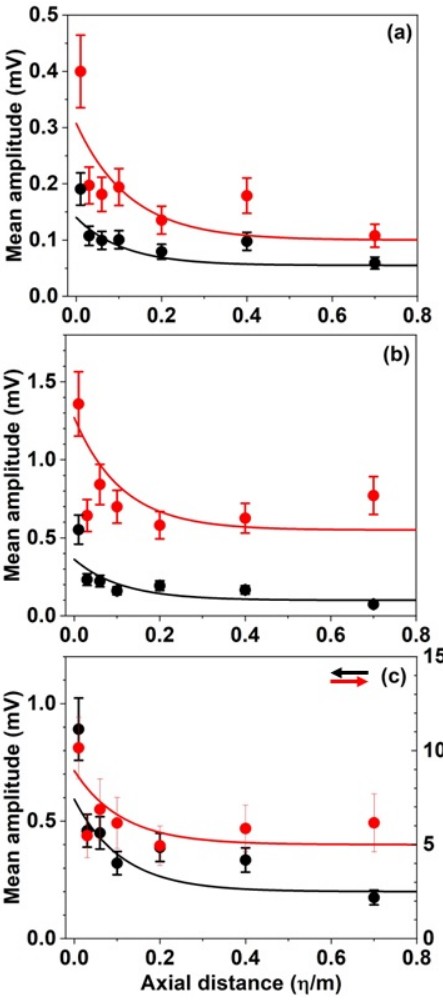

**Figure A5.** Observed mean amplitudes (solid circles) and simulated maximum $u_z$ displacements (lines): (**a**) D0.3P2η1 (black) and D0.3P4η1 (red), (**b**) D0.5P2η1 (black), D0.5P4η1 (red) and (**c**) D0.8P2η1 (black) and D0.8P4η1 (red). In the simulation, $\kappa_z = 9$, and FRS = 250 N/m$^2$ for P4 and 91 N/m$^2$ for P2. The simulated values were multiplied by $8.77 \times 10^7$ mV/m for D0.3P2 and D0.3P4, $2.70 \times 10^8$ mV/m for D0.5P2 and D0.5P2, and $4.05 \times 10^8$ mV/m for D0.8P2 and $1.47 \times 10^9$ mV/m for D1.2P4.

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
