# Peer review of "The Characteristics of Acoustic Emissions Due to Gas Leaks in Circular Cylinders: A Theoretical and Experimental Investigation"

_applsci, doi:10.3390/app13179814_

Round 1
Reviewer 1 Report
In this work, the authors present a study on analytical modeling for acoustic emission due to leakage through a circular pinhole in a gas storage cylinder. By solving the Navier-Lamé equation, they derived displacement fields responsible for acoustic emission and the concentrated force associated with the turbulent flow through the pinhole. Moreover, they performed a series of experiments to obtain the characteristics of the acoustic emission signals created by the leaks in a gas cylinder. The results confirmed the accuracy of the simulation results. The paper’s results are exciting and can be used in different areas of engineering.
However, before I can recommend the work for publication, the authors should revise their manuscript.
First, a better description of the difference between the previous analytical results (Ref. 18) and the presented analysis in this work. The authors should justify the need to repeat the cylindrical coordinates' analytical derivations.
Second, the relation between the diameter and the parameter kz should be explained better since it seems to be important for the frequency band of the emission.
Minor comments: The text should be revised for typos. For instance, in the legend of Fig. 8.
Finally, the authors should improve the connections between sentences and paragraphs.
Authors should improve the clarity of the text, as well as the connections between distinct parts of the paper.
Reviewer 2 Report
1.Figure 3 lacks a legend.
2.The frequency of sampling the acoustic emission signal is not indicated. Does this influence the results of the experimental analysis? Please explain.
3.If possible, please add the reference of every equation.
4.The conclusion should be concise and have a clear structure.
Author Response
1.Figure 3 lacks a legend.
--> For clarity, the condition for detecting AE signal has been supplemented in lines 227 – 229.
2.The frequency of sampling the acoustic emission signal is not indicated. Does this influence the results of the experimental analysis? Please explain.
--> As described in lines 227 – 229, a hit above the threshold of 40 dB was detected during maximum hit duration (MHD) of 1.0 ms. AE hits were collected during 300 s under the given experimental conditions. Prior to the main measurement, we investigated the effects of MHD and collection time on the observed AE parameters. We found that the 1.0 ms MHD and the 300 s collection time are long enough to not affect the results.
3.If possible, please add the reference of every equation.
--> Some references have been added.
4.The conclusion should be concise and have a clear structure.
--> There are only 132 words in the conclusion section. We have only changed the wording because reducing the size of the conclusion could make it unclear to understand the conclusion of the paper.
Reviewer 3 Report
This manuscript presents a theoretical and experimental investigation of acoustic emission originating from gas leaks in a cylindrical pressure vessel. Although acoustic emission (AE) has been widely used and studied as a nondestructive evaluation technique for detecting crack initiation and propagation, fiber breaks and matrix cracking (in composite materials) resulting from the stress waves generated from these damage events, AE is also generated from fluctuating Reynold's stresses (FRS) resulting from the interaction between a turbulent fluid flow and the solid structure. Although many experimental investigations investigating AE generated by gas leaks have been presented over the past three decades, a comprehensive theoretical model of AE excited by gas leaks in cylindrical structures has been lacking. This work fills in this knowledge gap by deriving the concentrated force-incorporated potentials (CFIP) that cause the FRS. The displacements associated with the longitudinal and shear components of the ultrasonic waves are analyzed. Experimental results from the leakage of nitrogen gas from an aperture in a steel gas cylinder are analyzed and compared with theoretical predictions. The angular and axial position dependencies of the AE signals generated by the pinhole stresses were found to have good agreement between theory and experiment. The results in this work are clearly presented. Given the relevance of the AE technique to the NDE community as a whole, I believe that this manuscript is suitable for publication in Applied Sciences.
A few points that need to be addressed:
1. In the experiment, why did the authors choose AE sensors with a resonant frequency of 150 kHz? Why were wideband sensors not used instead?
2. The locations of the sensors are not shown in Figure 2. Could the authors add (perhaps in an inset), the position and locations of the AE sensors on the test cyclinder?
3. Could the authors comment on the complexities associated with extending their model to elastically anisotropic materials, or composite pressure vessels?
The quality of the English language in the manuscript is very good. I did not find any grammatical errors or spelling mistakes.
